# Bio-Nanocomposite Based on Edible Gelatin Film as Active Packaging from *Clarias gariepinus* Fish Skin with the Addition of Cellulose Nanocrystalline and Nanopropolis

**DOI:** 10.3390/polym14183738

**Published:** 2022-09-07

**Authors:** Sri Aprilia, Nasrul Arahman, Muhammad Roil Bilad, Hazwani Suhaimi, Agus Arip Munawar, Indera Sakti Nasution

**Affiliations:** 1Doctoral Program, School of Engineering, Post Graduate Program, Universitas Syiah Kuala, Darussalam, Banda Aceh 23111, Indonesia; 2Department of Agricultural Engineering, Universitas Syiah Kuala, Darussalam, Banda Aceh 23111, Indonesia; 3Department of Chemical Engineering, Universitas Syiah Kuala, Darussalam, Banda Aceh 23111, Indonesia; 4Graduate School of Environmental Management, Universitas Syiah Kuala, Darussalam, Banda Aceh 23111, Indonesia; 5Research Center for Environmental and Natural Resources, Universitas Syiah Kuala, Jl. Hamzah Fansuri, No. 4, Darussalam, Banda Aceh 23111, Indonesia; 6Atsiri Research Center, Universitas Syiah Kuala, Jl. Syeh A. Rauf, Darussalam, Banda Aceh 23111, Indonesia; 7Faculty of Integrated Technologies, Universiti Brunei Darussalam, Jalan Tungku Link, Gadong BE1410, Brunei

**Keywords:** catfish skin nanogelatin, cellulose nanocrystal, nanopropolis, bio-nano composite, edible film

## Abstract

This study develops bio-nano composite gelatin-based edible film (NEF) by combining nanogelatin, cellulose nanocrystal (CNC), and nanopropolis (NP) fillers to improve the resulting film characteristics. The NEF was characterized in terms of thickness, swelling, pH, water content, solubility, vapor and oxygen permeability, mechanical properties, heat resistance, morphology, transparency, and color. The results showed that the thickness and swelling increased significantly, whilst the pH did not significantly differ in each treatment. The water content and the water solubility also showed no significant changes with loadings of both fillers. At the same time, vapor and oxygen permeability decreased with addition of the fillers but were not significantly affected by the loading amounts. The heat resistance properties increased with the filler addition. Tensile strength and Young’s modulus increased for the films loaded with >3% CNC. The elongation at break showed a significant difference together with transparency and color change. The greater the CNC concentration and NP loading were, the darker the resulting transparency and the color of the NEF. Overall results show a considerable improvement in the properties of the resulting NEFs with the incorporation of CNC and NP fillers.

## 1. Introduction

Synthetic plastic production is estimated to increase by approximately 320 million tons annually [1]. More than 40% of the produced plastic is used for packaging [2], with <14% eligible for recycling [3], thereby causing a bad impact on the environment [4]. Meanwhile, biodegradable films can be an alternative to synthetic polymers due to their edible, non-toxic, natural, and renewable properties [5,6,7,8,9]. Edible films can also be employed as functional component carriers in the production of bioactive packaging.

Packaging that actively alters the packed goods and extends their shelf life is identified as active biodegradable packaging [10,11]. Edible films can be made from gelatin (an animal protein [12,13,14] with active functions [15]. Gelatin-based films with good gas exchange barriers and strong mechanical qualities are increasingly employed in food packaging [16,17,18,19]. However, they have weak moisture barrier capabilities with significantly high sensitivity [20,21,22]. A few other properties support gelatin’s suitability as a biopolymer for the manufacturing of biodegradable packaging films [23,24,25,26,27] including edibility, biodegradability, excellent barrier capacity, and transparency.

Gelatin has hydrophilic-lipophilic properties and, when combined with polysaccharides, it can form films [28,29,30] and has been used in food packaging [31]. The presence of gelatin in a film form polymer matrix which significantly limits the diffusion of some active substance [32,33] and improve the biodegradability [34]. However, it is sensitive to moisture and has weak mechanical properties [35,36]. One approach to overcome these limitations is by incorporating other materials into the polymer matrix as composites or nanofillers [2]. Gelatin has been discovered to improve the mechanical resistance and barrier properties of bio-composite films [37]. Starch, cellulose, chitosan, and lipids are examples of composite film materials [38]. A mixture of the two polymers can improve the resulting film by interacting with various polysaccharide polymers [39]. Gelatin can form crosslinked networks in mixed systems, providing a robust and flexible film when suitably plasticized [40]. Combining different biopolymers is one such method for improving film characteristics [41].

Biodegradable active packaging is one of the solutions developed to improve the poor mechanical, thermal, and water barrier qualities of biopolymer-based films by incorporating nanofillers or nano reinforcements [42,43]. Bio-nanocomposites are hybrid systems consisting of biopolymers and fillers [44,45]. Recently, bio-nanocomposites made using nano-cellulose as filler have been developed [46,47,48,49]. Nanocrystal cellulose (CNC) strengthens the polymer matrix by forming a well-distributed percolation network through hydrogen bonds [50]. CNC is a polysaccharide-based nanomaterial found in polymers [37,51], and is used to improve the characteristics of the films, such as the combination of bio-nanocomposites and gelatin [52,53,54]. Bio-nanocomposite films can increase tensile strength and Young’s modulus [55,56]. The incorporation of nanoparticles into the polymer matrix increases barrier efficiency while lowering the costs and improving applicative performance [57].

Plasticizers are low-volatility molecules introduced to biopolymer materials. They adjust functional qualities, such as elasticity, stiffness, extensibility, mechanical properties, and dispensability [22]. Glycerol, sorbitol, ethylene glycol, vegetable oil, waxy lecithin, mono-, di-, and oligosaccharides, and amino acids are the most commonly used plasticizers in films [58]. Propolis is a resin-like material made by bees comprising of flavonoid resin, essential oil, wax, pollen, and other organics. The flavonoid fraction is a well-known substance with antibacterial property [59,60]. Moreover, propolis is a food-grade chemical previously utilized in edible films as an antibacterial agent [61]. It has been employed in film-infused formulations due to its antibiotic, antifungal, and antibacterial characteristics [62,63,64]. Crosslinked polymer nanoparticles with a hydrophobic core and hydrophilic shell are used to produce propolis nano-food [65]. A food packaging film with antimicrobial properties can act as an effective physical barrier against bacterial invasion and prolong the shelf life of foods [66]. Studies on the fabrication of gelatin-based bio-nanocomposites edible films (NEFs) have been reported. However, the use of a combination of CNC as the filler and nanopropolis (NP) as bioactive materials has not been reported.

This study reports the synthesis and characterization of a NEF. It was fabricated using the gelatin matrix obtained from *dumbo* catfish (*Clarias gariepinus*) skin, CNC filler from waste banana peduncles (*Musa acuminata* L.) and NP as the bioactive material under various compositions. The addition of NP imposed antibacterial properties. Hence, the prepared NEF was classified as active packaging. The characterization of the prepared NEFs included the thickness, water content and water solubility, swelling, pH, morphology, heat resistance, mechanical properties, water vapor and oxygen permeability, transparency, and color change.

## 2. Materials and Methods

### 2.1. Materials

Dumbo catfish (*Clarias gariepinus*) skin was obtained from local dumbo catfish farmers in Aceh Besar, Indonesia. The waste of Barangan banana peduncles was obtained from banana traders in Aceh Besar, Indonesia. The chemicals used were sulfuric acid (MilliporeSigma, Burlington, MA, USA), NaOH (MilliporeSigma, Burlington, MA, USA), H_2_O_2_ (35% food-grade), acetic acid (MilliporeSigma, Burlington, MA, USA), NaCl, carboxymethyl cellulose (CMC, PT. Multirasa, Bekasi, Indonesia), glycerol (Sigma-Aldrich, St. Louis, MO, USA), NP (CV. Elanazma Prima, Bekasi, Indonesia), and distilled water.

#### 2.1.1. Preparation of Cellulose Nanocrystal (CNC)

*Barangan* banana peduncles (*Musa acuminata* L.) were cut into pieces and pressed using a pressing machine to obtain the fiber. The alkalization process was performed using a microwave (Samsung ME731K) with a 1 M NaOH solution at 450-Watt power for 1 h. After polymerization, the bleaching was conducted for an hour using 35 wt% of H_2_O_2_ with a fiber-to-solution ratio of 1:15 (*w*/*v*) at 180 W microwave power and a temperature range of 60–65 °C. The CNC isolation procedure was carried out for 1.5 h by hydrolysis using 3 M sulfuric acid at a ratio of 1:15 (*w*/*v*) at 100 W microwave radiation under a temperature range of 45–52 °C. The sample was then rinsed with distilled water until a neutral pH was reached. Subsequently, the sample was centrifuged at 11,000 rpm for 30 min, and the water was changed twice, followed by the sonification for 4 h. It was then centrifuged for 30 min and dried at 60 °C until a maximum moisture content of 12% was reached before being crushed and sieved using a 500-mesh sieve according to ASTM No: 11. The prepared CNC had the following characteristics in terms of morphology, functional groups, crystallinity, heat resistance, particle size, and color. The FTIR spectra of CNC showed the most relevant molecular bands, namely O-H, C-H, and C-O, at the corresponding wavenumbers range of 3200–4000 cm^−1^, 2500–3200 cm^−1^, and 500–1500 cm^−1^, respectively. The thermogravimetric analysis showed that it decomposed at a temperature of 275.35–409.40 °C, with a weight loss range of 84–94%. The crystallinity index and crystal sizes ranged from 54% to 57% and 22.35% to 36.28 nm, respectively.

#### 2.1.2. Preparation of Dumbo Catfish Skin Nanogelatin (NGC)

Fresh *dumbo* catfish skin was soaked in water for 2 h, then cleaned and reduced in size. Next, it was soaked in 0.1 M NaCl solution for 6 h, followed by soaking in 0.2 M NaOH for 2 h, with a ratio of fish skin to a solution of 1:15 (*w*/*v*). The solution was changed every hour. The sample was then washed with distilled water until the neutral pH was achieved, followed by immersion in a 0.1 M acetic acid solution with a ratio of 1:15 (*w*/*v*) for 6 h. Subsequently, the fish skin sample was washed and extracted with distilled water in a ratio of 1:5 (*w*/*v*) for 4 h. The extraction was performed using a microwave (Samsung ME731K) at a power of 180 W in the temperature range of 60–65 °C. Next, the sample was dried at 50 °C to reach a maximum moisture content of 16%. The prepared NGC was characterized and posed the following properties. The proximate analysis resulted in protein, fat, ash, and water content of 71.69%, 9.33%, 1.42%, and 17.56%. The pH and viscosity of the NGC solution were 7.1–7.2 and 1.25–6 cP. The FTIR analysis identified dominant peaks at 2500–4000 cm^−1^ (amide A: OH and CH stretching), 1500–2000 cm^−1^ (C=O of alkene), and 500–1500 cm^−1^ (C-N of methylene). The NGC sample decomposed at a temperature range of 267.63–427.23 °C mainly due to the decomposition or degradation of organic substances, while the decrease in the NGC mass occurred until a temperature of 600 °C resulting in a residual fraction of 15.42–20.17%. The XRD peaks showed 2 θ ranging from 7.33° to 9.68°, which was <20°, indicating the amorphous structure, with a crystallinity range of 9–64%. The average particle size of the NGC was 236.2 nm.

#### 2.1.3. Preparation of Bio-Nanocomposites Edible Film

Solutions containing 1% (*w*/*v*) of gelatin (NGC), 1% of CMC (*w*/*v*) and the 1%, 3%, and 5% (*w*/*w*) of CNC in distilled water were prepared by stirred for 1.5 h under temperature of 60 °C. The NGC, CNC, and CMC solutions were then mixed, reheated on a hot plate at 60 °C for 30 min, then cooled. Subsequently, it was loaded with glycerol 0.25% (*v*/*v*) and NP 0.1% (*w*/*v*) by weight of the NGC under continuous stirring. The treatment design is shown in Table 1 and the study diagram is presented in Figure 1. The solution was then casted in a 20.5 cm × 18.5 cm mold and dried in an oven at 50 °C for 20 h. Then, it was released from the mold to form an edible film. All samples were prepared in triplicate. In this study, carboxymethyl cellulose was added as a stabilizer. Based on the results of the earlier trial, film formation was found to be difficult, and when formed it was very sticky.

### 2.2. Characterization of NEF

#### 2.2.1. Thickness

After taking the prepared NEF from the mold, its thickness was then measured using a screw micrometer (Model Mitutoyo, Kanagawa, Japan) at four random points. The thickness data are presented as average ± standard deviation.

#### 2.2.2. Water Content and Water Solubility

An empty 50 mL bottle was coded and weighed as the initial weight (W1). Then, a 30 mm × 30 mm NEF sample was placed in the bottle and weighed (W2). The NEF was subsequently dried at 105 °C for 24 h and weighed (W3). Afterward, 15 mL of distilled water was added to the bottle containing the NEF and stirred. It was left for 24 h, and the water was drained. This was followed by drying at 105 °C for 24 h prior to weighing (W4). The water content (WC) and the water solubility (WS) were calculated using Equations (1) and (2) [27].
(1)$$\mathrm{WC}\;(\%)=\frac{W2-W3}{W2-W1}\times 100\%$$
(2)$$\mathrm{WS}\;(\%)=\frac{W3-W4}{W3-W1}\times 100\%$$

#### 2.2.3. Swelling

A 30 mm × 30 mm NEF sample was weighed as dry weight (Wi) and immersed in distilled water for 10 min. The film was removed, and the remaining water was also removed with a filter paper before being reweighed to obtain the wet weight (Wf). The degree of swelling was calculated using Equation (3) [3].
(3)$$\mathrm{Swelling}\;(\%)=\frac{Wf-Wi}{Wi}\times 100\%$$

#### 2.2.4. pH

A 30 mm × 30 mm NEF sample was immersed in distilled water for 10 min, then it was removed, and the pH of the water was measured using a pH-meter.

#### 2.2.5. Mechanical Property

A NEF specimen was prepared and tested according to the ASTM D 638-03 method using a Universal Testing Machine (Type HT-8503, Seremban, Negeri Sembilan, Malaysia). The mechanical properties of the NEF samples were tested in terms of its tensile strength (TS), elongation at break (EB), and Young’s modulus (YM).

#### 2.2.6. Morphology

The morphology of NEF samples was assessed by scanning electron microscopy (SEM). The samples were observed using SEM (JSM-6510LA, JEOL, Akishima, Tokyo, Japan) with 10-k magnification at 30 kV power. After coating, the specimen was mounted and when the height of the surface is the same as that of the holder, the SEM machine was turned on and scanning was carried out.

#### 2.2.7. Water Vapor Permeability (WVP)

A 9 mL bottle was filled with 6 mL of distilled water and covered with a 30 mm × 30 mm NEF. The bottle was then weighed and placed in a desiccator at room temperature. This was carried out every 24 h for three days, while the WVP (in g m^−1^ Pa^−1^ h^−1^) of the NEF was calculated using Equation (4) [27].
(4)$$\mathrm{WVP}=\frac{\Delta m\times d}{A\times t\times \Delta P}$$
where d is the film thickness (mm), m is the mass loss (g), A is the bottle opening area (m^2^), t is the measurement time interval (h), and *P* (about 4247 Pa) is the difference in partial vapor pressure between the two sides of the film at room temperature and 100% relative humidity.

#### 2.2.8. Oxygen Permeability 

The oxygen permeability (OP) was measured using a procedure reported earlier [37] with slight modifications. Gelatin-based films were cut into 30 mm × 30 mm, while the 9 mL bottle was covered with a NEF and weighed as the initial weight before being placed in a desiccator. The bottle was also weighed every day for three days, while the oxygen permeability transmission rate (OPTR) and oxygen permeability (OP) were calculated using Equations (5) and (6) [37].
(5)$$\mathrm{OPTR}=\frac{Slop}{Film\;area}$$
(6)$$\mathrm{OP}=\frac{OPTR\times L}{\Delta P}$$
where ΔP is the difference in partial vapor pressure between pure water and dry air (0.02308 atm/1.013 × 10^5^ Pa at 25 °C), while L (m) is the average film thickness.

#### 2.2.9. Thermal Stability

Heat stability properties of NEF samples were analyzed using differential scanning calorimetry (Shimadzu DSC-60). A 3 mg sample was placed into an aluminium pan and then scanned at a temperature range from 30 °C to 600 °C at a scan rate of 40 °C per minute under nitrogen atmosphere at a constant flow rate of 20 mL per minute.

#### 2.2.10. Transparency

Transmittance refers to the fraction of incident light that is transmitted. Transparency is the condition of being see-through, hence can be quantified using the transmittance. Light transmission of NEF samples was measured by a UV-Vis Spectrophotometer (Shimadzu Spectrophotometer UV-1800, Kyoto, Japan). The sample of 4 cm × 5 cm was subjected to measurement at a wavelength range of 200–800 nm. The transparency of the NEF sample was calculated according to earlier work [67] using Equation (7), where *T*_600_ is the percent transmittance at 600 nm and *L* (m) is the average film thickness.
(7)$$\mathrm{Transparency}\;({\mathrm{mm}}^{-1})=\frac{-\mathrm{Log}\left(T_{600}\right)}{L}$$

#### 2.2.11. Color

CIELAB is the most comprehensive $L^{*}a^{*}b^{*}$ color space defined by the International Commission on color illumination (French Commission Internationale de l’eclairage, CIELAB). It can describe visible colors, and is frequently used for reference color spaces [68]. The color analysis was performed using licensed MVtec Halcon ver.20 software. The background was separated from the original image, and then each sample object was separated by giving 15 times erosion treatment to obtain the RGB value. The RGB (Red, Green, Blue) values formed were converted to the $L^{*}a^{*}b^{*}\;$ color space. The NEF sample was placed on a white plate, and the values were calculated as a standard. The color difference ($\Delta E^{*}$) was calculated using Equation (8) [51].
(8)$$\Delta E^{*}=\sqrt{{\left(\Delta L^{*}\right)}^{2}+{\left(\Delta a^{*}\right)}^{2}+{\left(\Delta b^{*}\right)}^{2}}$$
where $\Delta L^{*}$, $\Delta a^{*},$ and $\Delta b^{*}$ are the color differences between the NEF sample and the standard white ($L^{*}$ = 90.6153, $a^{*}$ = 1.69261, and $b^{*}$ = 0.5891).

### 2.3. Statistical Analysis

The test data obtained were in the mean ± standard deviation and were further analyzed using One-way ANOVA (Analysis of variance). Duncan’s test was also carried out at a significant level of 5%. All data including three repetitions were analyzed using SPSS ver. 24.

## 3. Results and Discussion

### 3.1. Thickness of NEF

Figure 2a shows that the resulting NEF had thicknesses ranged from 0.138 to 0.164 mm. The NEF turned thicker when composited with CNC and NP due to the increase in the solids content of the sample. The ANOVA confirmed that the bio-nanocomposite had a significant effect with * *p* < 0.05 on the NEF thickness. Figure 2 shows superscripts with the same letter notation indicating no significant differences in Duncan’s test *α* = 0.05. A significant difference was observed between NGC before and after being composited with CNC and NP, namely between the NCC00 sample and the rest. The NEF’s thickness was further increased when composited by NP, as can be seen by comparing NCC01 vs. NCC04, NCC02 vs. NCC05, and NCC03 vs. NCC06, in which the increments of the first two pairs were significant. The film thickness tends to increase with a rising dry weight of the material for filmmaking [13]. This result is in line with the work of Khodaei and Hamidi [4], who reported that the thickness of the gelatin film increased at higher loading of the composite material.

### 3.2. Swelling of NEF

The swelling of the NEF showed an increasing ratio when composited with CNC and NP as shown in Figure 2a with the values ranging from 17.0% to 28.8%. Figure 2a shows superscripts with the same letter, indicating no significant differences in the Duncan’s test (α = 0.05) between NCC00 and NCC01. However, as the concentration of cellulose nanocrystal increased from 0% to 3%, the swelling ration increased significantly. Beyond that, NCC02, NCC03, NCC04, and NCC05 showed no significant increase in the swelling. A significant increase in the swelling could also be seen between NCC05 and NCC06, in which the cellulose nanocrystal increased from 3% to 5% in addition of 0.1% of NP. Generally, loading more solid into the NEF resulted in an increase in swelling. The higher the concentration of CNC, the greater the swelling. This increase might be due to its ability to bind with water. The ANOVA results showed that edible film bio-nanocomposites had a significant effect with ** *p* < 0.01 on the swelling of the NEF.

The NCC00 sample was not statistically significantly different from NCC01 but differed from the other samples. Meanwhile, the NCC02 sample was not significantly different from NCC03, NCC04, and NCC05, but NCC06 was significantly different from all other treatments. Materials with hydrophilic properties will easily bind with water, causing large swelling [37]. In contrast to hydrophobic materials, which are more difficult to bind with water, the swelling of the film is also small. The branched polysaccharide structure might cause large swelling of the film [4].

### 3.3. Water Content

Figure 2b shows that water content of the prepared NEFs ranged from 19.0% to 22.3%. The lowest water content was NCC06 sample with 19.0%. The ANOVA results showed that the water content of NEFs was not significantly affected by the solid loadings as demonstrated by *p*-values > 0.05. As suggested from an earlier report, the water content of NEF is not affected by the concentration of CNC and the addition of NP, but it was significantly influenced by the temperature and drying time. A previous study found that the water content of gelatin-based bio-nanocomposite films ranged from 11.0% to 29.1% [45].

### 3.4. Water Solubility

An important property of edible films is their behavior in contact with water molecules [48]. Biodegradable films must be protective for food products but decompose when released into the environment [67]. This implies that the water solubility of the film must be measured to determine its capacity to remain intact during storage but subsequently diminish. The water solubility of the NEFs ranged from 34.363% to 39.269% as shown in Figure 2b. Based on the results, the highest value was found for the NCC00 sample with 39.3%, which was not supplemented with either CNC or NP. The remaining NEFs that contained CNC and/or NP had lower water solubility by 4.9%. The ANOVA results show that loadings of CNC and NP was insignificant (*p*-values > 0.05) on the water solubility. The finding shows that the addition of CNC filler reduced the water solubility of the resulting NEFs as seen by comparing water solubility data of NCC00 (with superscript b) to the rests (with superscript ab and b). Given that gelatin is hydrophilic, adding polysaccharide fillers, such as cellulose, reinforces the protein layer and increases its resistance to moisture [56]. The most optimal CNC loading to produce good water resistance and mechanical properties for strengthening the gelatin-based film was reported at 5% *w*/*w* [69].

The range of water solubility of NEF reported in literature is 35.9–52.9%. According to Tessaro et al. [45], CNC as a filler not only improves the mechanical properties of the film but also the moisture resistance and provides stability in high humidity environments. Oyeoka et al. [56] added that the water absorption was 22.50% with the addition of 5% CNC, but it was reduced to 19.18% and 19.05% when the CNC loading was increased to 10%. Moreover, water absorption decreased from 13.8% to 10% when 10% CNC was added to the starch-based film. In another study, the addition to starch-based bio-nanocomposite films reduced water absorption from 17.0% to 15.7%. This reduction can be ascribed to the development of hydrogen bonds between the CNC and the matrix. CNC can inhibit the formation of cavities by this connection, making the penetration of moisture impossible.

### 3.5. pH of NEF

The pH values of the prepared NEFs are shown in Table 2 where the loading of CNC and NP did not affect the pH of the resulting films with *p*-values > 0.05 from the ANOVA. Tessaro et al. [45] reported a strong interaction between the negative CNC and the functional group that is positively charged. In contrast, B-type gelatin is a polyelectrolyte whose primary charge is affected by the pH of the solution. Gelatin also has a higher affinity for water, which might be unaffected by the presence of nanocrystals.

### 3.6. Water Vapor Permeability

Overall, the WVP of NEF decreased after adding CNC and NP fillers, as shown in Table 2. The barrier property of edible films is an important factor for controlling moisture transfer and prolonging the shelf life of food products [28]. A lower WVP value suggests a better water vapor barrier. The ANOVA results showed significant effects of both CNC and NP loadings on the NEF’s WVP (*p* < 0.05). When loaded with CNC (NCC01, NCC02 and NCC03; all labelled with superscript b), the NEF had the WVPs of 2.9 ± 0.1 × 10^−7^ to 3.0 ± 0.2 g^−1^ m^−1^ Pa^−1^ h^−1^, which is significantly higher than the WVP of NCC00 (without CNC, labelled with superscript b). Introducing the NP to the film increased the WVP of the CNC containing film to a range of 3.0 ± 0.3 × 10^−7^ to 3.4 ± 0.4 g^−1^ m^−1^ Pa^−1^ h^−1^, but still lower than the WVP of NCC00. According to Azizi-Lalabadi et al. [67], the first decrease in WVP is probably due to CNC nanoparticles filling the pores and connecting the polymer chains, which reduce the number and size of pores in the film. Furthermore, the nanoparticles might function as physical barriers, increasing the distance of the diffusing water molecules due to pore bending and curves. This phenomenon is also known as tortuosity.

Table 2 shows that loading the CNC significantly reduced the WVP value, as also reported elsewhere [70], but the CNC concentration did not significantly affect the WVP. However, when the NP was added, the value of the WVP increased again (see Samples NCC04, 05 and 06). This can be ascribed to the elevated affinity of hydroxyl group to bind with water. Similar to the CNC loading, the concentration of NP did not significantly affect the WVP. The hydroxyl groups of CNC filled the space between the gelatin chains, leading to the formation of a dense network, which in turn increases the water resistance. In the presence of NP, the free hydrophilic groups were decreased due to hydrogen bonds between the gelatin and the NP in the gelatin structure [42], thereby increasing the WVP.

### 3.7. Oxygen Permeability

The OP of the NEF is directly proportional to the OPTR, as shown in Table 2. The OPTR, and OP values showed the same trend and ranged from 0.9–3.1 g m^−1^ d^−1^ to 1.4 × 10^−9^–4.6 × 10^−9^ g^−1^ m^−1^ Pa^−1^ d^−1^, respectively. The highest values of the OPTR and OP were found in the NCC00 sample with 3.1 g m^−1^ d^−1^ and 4.6 × 10^−9^ g^−1^ m^−1^ Pa^−1^ d^−1^, while the lowest was in the NCC05 with 0.9 g^−1^ m^−1^ d^−1^ and 1.4 × 10^−9^ g^−1^ m^−1^ Pa^−1^ d^−1^, respectively. The low values of OPTR and OP are desirable for the NEF, suggesting the effectiveness of the CNC and NP loading in improving the resulting film properties. Loading of the CNC significantly reduced both OPTR and OP, but the effect of the CNC concentration within the tested range was in significant. Similarly, further loading of 0.1% of NP on CNC containing NEFs lowered both the OPTR and OP values, but only significantly at CNC concentrations of 3% and 5% (NCC005 and NCC006).

As the concentration of CNC or NP in the NEF increased, the OP first decreased, then increased again. According to Azizi-Lalabadi et al. [67], the first decrease in the OP can be attributed to the nanoparticles’ capacity in lowering the number and the size of pores in the polymer network. When the particle loading was further increased, the enhanced OP might be due to its interaction with the polymer network, allowing oxygen molecules to travel more freely. Leite et al. [11] stated that the WVP and OPTR are barrier qualities which define a bio-based film’s capacity to protect food products against moisture and oxygen transfer, lipid oxidation, volatile aroma, and taste loss.

When compared to literature, Ge et al. [1] reported that the use of gelatin bio-nanocomposite culminated in the OP and the WVP values ranged from 2.035 cm^3^ m^−2^ d^−1^ atm^−1^ to 1.323 cm^3^ m^−2^ d^−1^ atm^−1^ as well as 7.8 × 10^−4^ to 6.7 × 10^−4^ g day^−1^ m^−1^ atm^−1^. In addition, Yadav et al. [37] found that the OP value of composite gelatin-kitosan films ranged from 3.582 to 7.0347 × 10^−6^ cm^3^ m^−1^ d^−1^ atm^−1^.

### 3.8. Mechanical Properties of NEF

The mechanical properties of the TS, EB, and YM of the NEF samples are shown in Figure 3. The TS, EB, and YM values ranged from 114.67 to 164.91 MPa, 13.89% to 41.07%, and 380.00 to 1203.72 MPa, respectively. The ANOVA results showed no significant effects on TS with *p* > 0.05, but a significant difference * *p* < 0.05 was found on EB and MY. The data show no significant impact of CNC and NP loadings on the TS (Figure 3a). According to a previous study, the TS was lower when no bioactive agents was added [71]. The change in the mechanical characteristics of the film including polysaccharide and protein is presumably caused by the interaction between the two biopolymers. The inclusion of bioactive compounds raised the TS of all films while decreasing their EB, but the change was statistically insignificant. A previous study reported that TS decreased when the CNC concentration was increased from 5% to 10% [56]. This can be caused by a less-than-perfect dispersion. The aggregate particles are mostly responsible for the poor dispersion, causing stress points in the matrix and reducing CNC’s positive contribution to TS. Another study also suggested that optimizing the type and level of nanoparticles introduced to the system can improve the mechanical strength of composite polymers [67]. Furthermore, Ge et al. [14] reported that gelatin bio-nanocomposite TS values ranged from 2.35 to 9.44 MPa, while Yadav et al. [37] found that the value of the gelatin-chitosan composite film was between 10.54–17.11 MPa. This study used bio-nanocomposite edible film gelatin from *dumbo* catfish skin which produced a much higher TS value, ranging from 114.67 to 164.91 MPa.

The EB value generally decreased significantly when the CNC filler was added. The EB-value of NCC00 was significantly higher than the EB-values NCC01, NCC02 and NCC03. When those NEF containing CNC samples was loaded with NP, their EB-values decreased, but only the NCC04 sample showed a significant increase (Figure 3b).

Loading the CNC into the NEF generally increased the YM but only significant for the one with 3% loading. Loading CNC increased the YM-value, but the loading amount did not show any clear trend (Figure 3c). The maximum YM was observed from for NCC02, which was significantly higher than the rest. Loading NP to CNC containing film generally reduced the YM as seen by comparing NCC01 vs. NCC04, NCC02 vs. NCC05, and NCC03 vs. NCC06. The trend can be attributed to the following. The CNC loading promoted crosslinks between the polymer molecules, thereby enhancing the YM of the composite films. Furthermore, the nanoparticles can cover parts of the polymer network’s holes, lowering their compressibility. They can potentially improve the mechanical strength of polymer films by acting as active fillers, as suggested by Azizi-Lalabadi et al. [67]. The presence of NP reduced the CNC impact on the YM property. The reinforcement of the films prepared in this work by CNC and NP was somewhat less than what reported earlier. Generally, all type of cellulose nanocomposite offer somewhat reinforcement to the resulting composite [72]. Nonetheless, it is worth noting that it is hard to make a fair and direct comparison with other studies because of the difference in the applied CNC and NP materials. We believe that the weak reinforcing ability of the NEF in this study compared to the ones reporter earlier was due to the properties of the CNC and NP fillers.

### 3.9. Thermal Resistance Properties of NEF

The thermal resistance property of the prepared NEF samples is depicted from the DSC shown in Figure 4. Table 3 summarized the glass transition (Tg), and the melting temperature (Tm), as well as the enthalpy which ranged from 127.76–151.39 °C, 304.40–354.68 °C, and 16.76 × 10^3^–77.19 × 10^3^ J/kg, respectively. The Tg value increased after the addition of both CNC and NP fillers. However, the Tm value increased after adding CNC filler, but decreased when further loaded with the NP filler, while the enthalpy showed an irregular trend. The findings suggest that the addition of CNC and NP affected the thermal properties of NEF. The highest Tg, Tm, and enthalpy values were obtained in samples NCC03, NCC02, and NCC06 with values of 151.39 °C, 354.68 °C and 77.19 × 10^3^ J/kg respectively. CNC has an anti-plasticizing impact on gelatin-based films due to the ultimate absorption of water and glycerol. However, it also acts as a reinforcement of the biopolymer matrix, reducing macromolecule mobility and contributing to an increase in Tg as reported earlier [45]. The addition of nano-complexes can increase the thermal stability of gelatin-based films. In nano-complexes, there are electrostatic interactions that offer enhanced thermal stability [16]. Recent studies also demonstrated that gelatin-based film formulations containing phenolics can build a strong gelatin network with the phenolic molecules, capable of raising the denaturation temperature and improving the thermal durability of the films.

### 3.10. Morphological Properties of NEF

The morphology of the NEF’s surface can be seen from the SEM images presented in Figure 5. Two images for each sample were provided for each sample with 200× magnification that show general view of the surface and 1000× magnifications that show more details of the morphology. A flat and smooth surface is shown for NCC00 sample. A white lump can be attributed to impurities stuck on the film surface. On the other hand, many flat surfaces without any crack or defect with some white lumps were spotted on top of films containing CNC filler. Those white lumps can be attributed to the aggregate of particle clumped together. The morphology shown by the SEM image was in line with other properties discussed earlier.

Figure 6 shows that the addition of CNC and NP caused the NEF samples to be opaquer. The gelatin film presented a smooth and continuous surface due to the homogeneous tissue structure. The addition of CNC into the polymer matrix caused a less smooth surface, indicating a microstructural change. Meanwhile, the existence of white dots can be attributed to the formation of some tiny cellulose nanocrystal aggregates and the enhanced nanocrystal cross-section in the polymer matrix [73]. The surface of the pure gelatin film was relatively smooth and consistent, indicating compatibility as reported in an earlier study [48]. However, minor fractures and folds were observed spreading on the gelatin-chitosan film. Plasticizers have numerous hydroxyl groups that can create intermolecular hydrogen bonds with the polymer matrix’s hydroxyl groups. The hydrogen bonding between the polymer and the matrix contributes to the films’ remarkable compactness and cohesion. When the amount of nanofiller applied exceeds the limit of uniform distribution and strengthening effect, aggregation occurs due to the increasing interaction between fillers [14]. The higher the concentration of the nano-cellulose was, the greater the aggregation behavior during water evaporation. A previous study stated that the aggregation of the nanofiller culminated in an unanticipated mechanical weakness of the composite film [47], in accordance with the decreasing EB-values obtained in this study.

### 3.11. Transparency Properties of NEF

The transparencies of the NEF samples are presented in Table 4. The transparency property is one of the most important characteristics in producing films for packaging specific food since it potentially prevents or slows down the oxidation of lipids, pigments, proteins, or vitamins. This feature directly affects food shelf life by minimizing unpleasant taste, color, odor, and nutrient loss, hence preserving the organoleptic and nutritional aspects of packaged goods [53].

Table 4 shows that the NEF’s transmittance decreased in the presence of CNC while the transparency increased. Higher CNC concentration lowered the transmittance and increased the transparency. This is true at different wavelengths employed in this study. Conversely, the transmittance decreased in the presence of the NP while the transparency increased. The increase in transparency is in line with the increasing concentration of CNC (for NEF samples without NP) and the addition of NP. The ANOVA results showed that the bio-nanocomposites edible film treatment had a significant effect on the transmittance and transparency values with ** *p* < 0.01. The highest transparency was found in NCC06 with 14.48 mm^−1^, while the lowest was in the NCC00 with 11.04 mm^−1^. The trend of transmittance and transparency values is in line with Figure 6. Therefore, it can be concluded that, in general, the transmittance value at various wavelengths shows the same trend as WVP, OPTR, and OP edible film NEF.

It was suspected that the NP contained phenolic compounds that contributed to the low NEF’s transparency values. As the concentration of particles in the NEF increased, its transparency decreased due to the rise in light scattering [67]. However, bioactive compounds present in the film, particularly phenolics, might also increase barrier properties, because they can absorb UV-Vis light. The unsaturated double bonds conjugated into saturated covalent bonds in some phenolic compounds were reported to be responsible for UV-Vis radiation absorption [45].

### 3.12. Color of NEF

The brightness and the color change of the prepared NEF samples are summarized in Table 5. The smallest Δ*L** parameter value was obtained in NCC00 at 16.079 indicating the brightest NEF color. There was an increase in Δ*L** when CNC was added in different concentrations together with NP. In contrast, the Δ*a** and Δ*b** values decreased with the darkening of the film color. The Δ*E** color difference is directly proportional to the parameter of Δ*L**, while the trend of color parameter for Δ*L**, Δ*a**, Δ*b**, and Δ*E** corresponds to the transparency when loaded with the CNC and the NP. The color and transparency results are consistent when compared visually with pictures in Figure 6. The ANOVA results also showed that the NEF had a significant effect on the color as demonstrated by ** *p*-values < 0.01. The segregation of color parameters into $\Delta L^{*}$, $\Delta a^{*}$, and $\Delta b^{*}$ was performed by MVtec Halcon ver.20 software. Those parameters indicate the deviation from the standard white. At this point, we have no confidence in analyzing the individual parameters. The most important parameter that can be associated with physical appearance (Figure 5) is $\Delta E^{*}$. It shows that loading CNC increased the color significantly but was not affected by the loading quantity (Table 5). Further loading the NEF composite with NP increased the color significantly, in which the color of NCC004 and NCC005 was more prominence than NCC006. This unclear trend suggests that there was interaction between CNC and NP that affect the NEF’s color. This interaction cannot be explained with available data in this study.

## 4. Conclusions

Overall results demonstrated changes of the resulting NEF properties when loaded with the CNC and the NP. The thickness of the NEF increased significantly with increasing CNC concentration and NP addition, with the value ranging from 0.138 to 0.164 mm. Higher CNC concentration led to increased swelling of the resulting NEF ranged at 17.0–28.8%. The CNC and NP loading did not affect the pH, whilst the water content, the water solubility, and WVP were in the ranges of 19.0–22.3% and 34.4–39.3% and 2.9 × 10^−7^ to 3.6 × 10^−7^ g m^−1^ Pa^−1^ h^−1^ respectively. Loading both CNC and NP lowered the water solubility by 4.9% for all treatments. High CNC concentration lowered the WVP value but increased it in the presence of NP. The OPTR and the OP were correlated linearly and ranged between 0.9 and 3.1 g m^−1^ d^−1^ and 1.4 × 10^−9^ and 4.6 × 10^−9^ g m^−1^ Pa^−1^ d^−1^, respectively. The TS, EB, and YM values ranged from 114.67 to 164.91 MPa, 13.89 to 41.07%, and 380.00 to 1203.72 MPa, respectively, in which a significant increase in TS and YM values was observed for NEF containing 3% of CNC. Moreover, the Tg, Tm, and enthalpy ranged from 127.76 to 151.39 °C, 304.40 to 354.68 °C, and 16.76 × 10^3^ to 77.19 × 10^3^ J/kg, respectively. The Tg value increased after adding CNC and NP, while the Tm increased when loaded with CNC but decreased in the presence of NP. The NEF’s light transmission decreased in the presence of CNC while the transparency increased. The value of Δ*L** increased when loaded with both the CNC and the NP. A future work on evaluation of the antimicrobial properties in protection against bacterial invasion and prolonging the shelf life of food is required to measure the additional benefit of NP filler on the NEF.

## Figures and Tables

**Figure 1 polymers-14-03738-f001:**
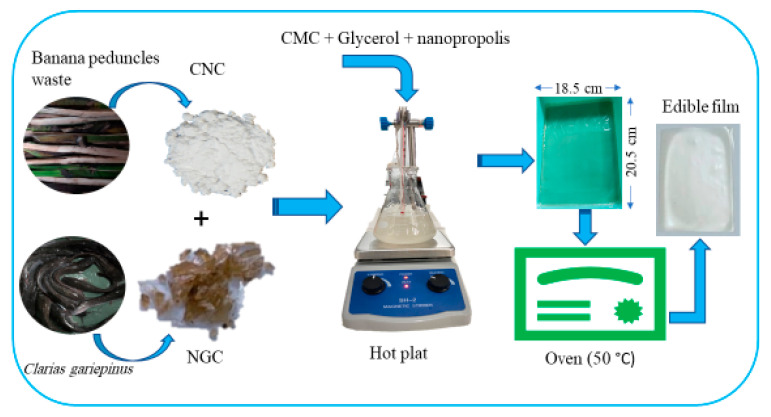
The illustration of the edible film fabrication process applied in this study.

**Figure 2 polymers-14-03738-f002:**
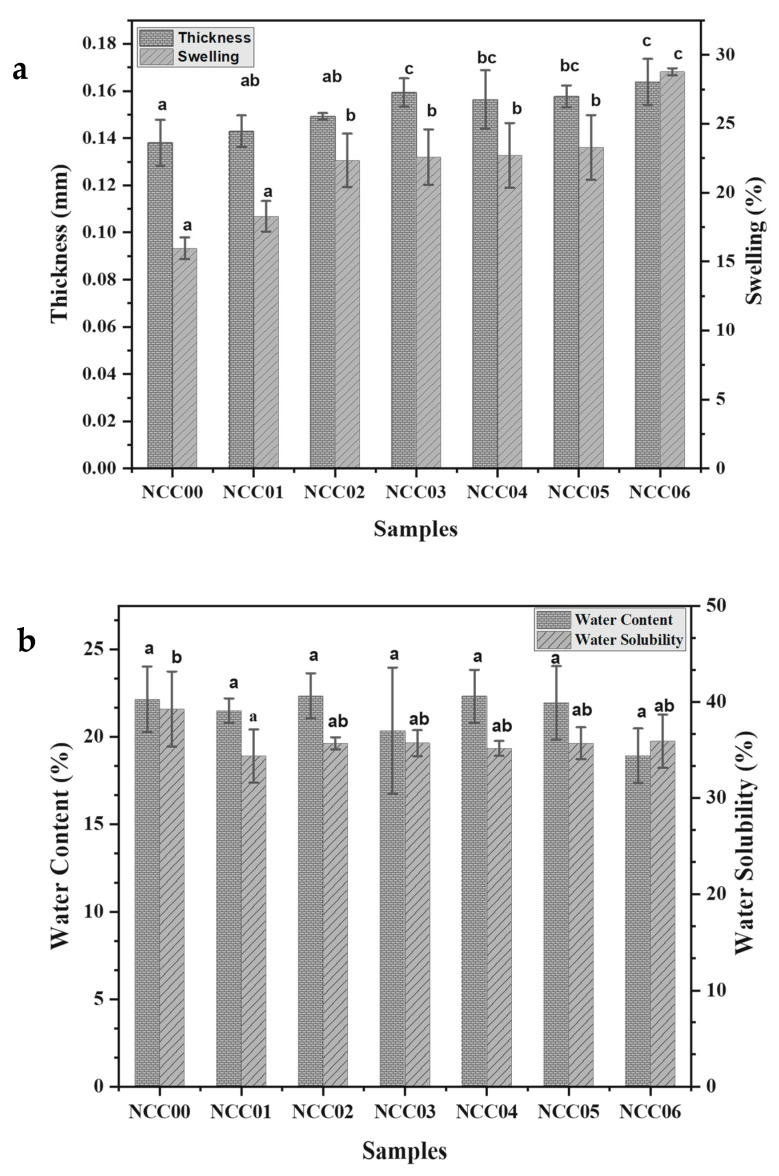
(**a**) The thickness and swelling of NEF as function of solid loadings; (**b**) The water content and the water solubility of NEF. Superscripts with the same letter show no significant differences in Duncan’s test.

**Figure 3 polymers-14-03738-f003:**
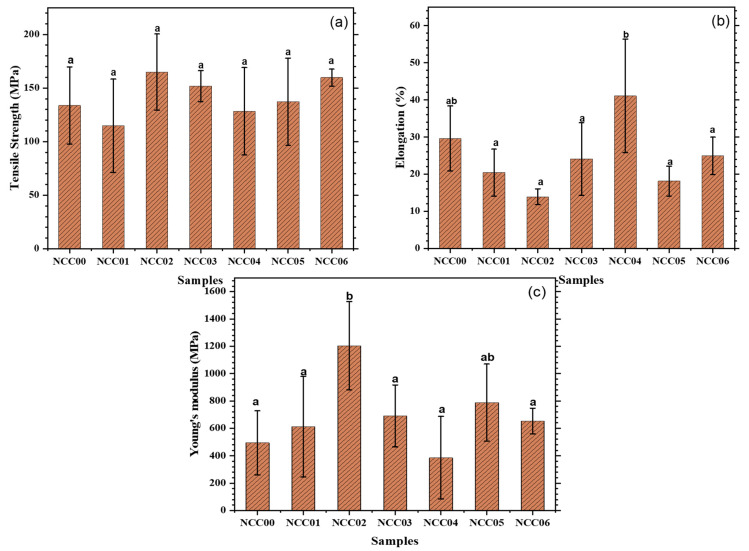
Values of (**a**) tensile strength; (**b**) elongation; and (**c**) young’s modulus. Superscripts with the same letter show no significant differences in the Duncan’s test (α = 0.05).

**Figure 4 polymers-14-03738-f004:**
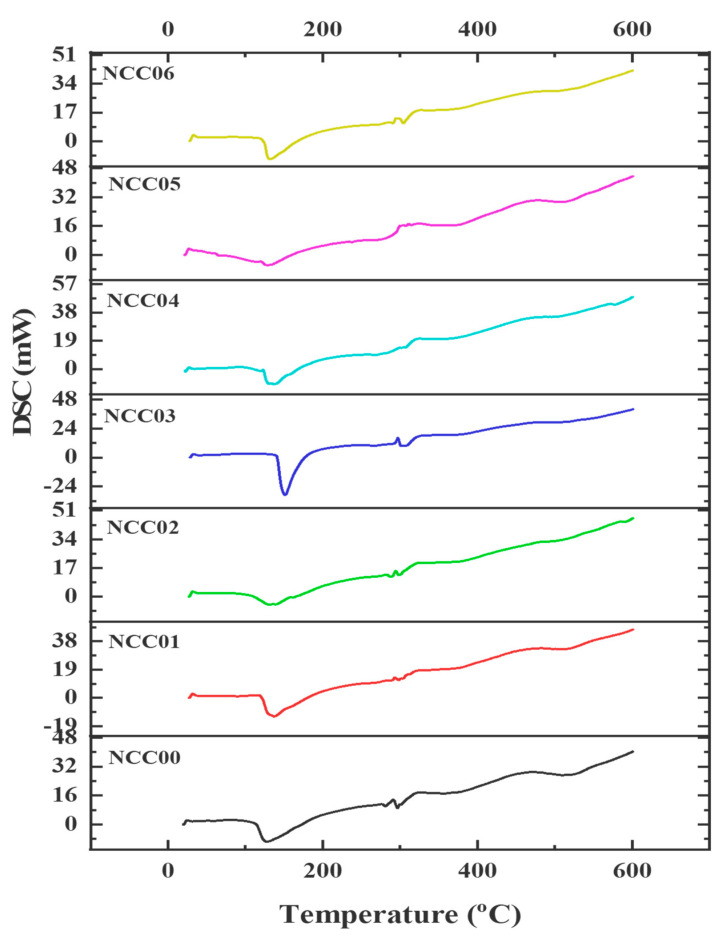
Curve thermal DSC sample NEF.

**Figure 5 polymers-14-03738-f005:**
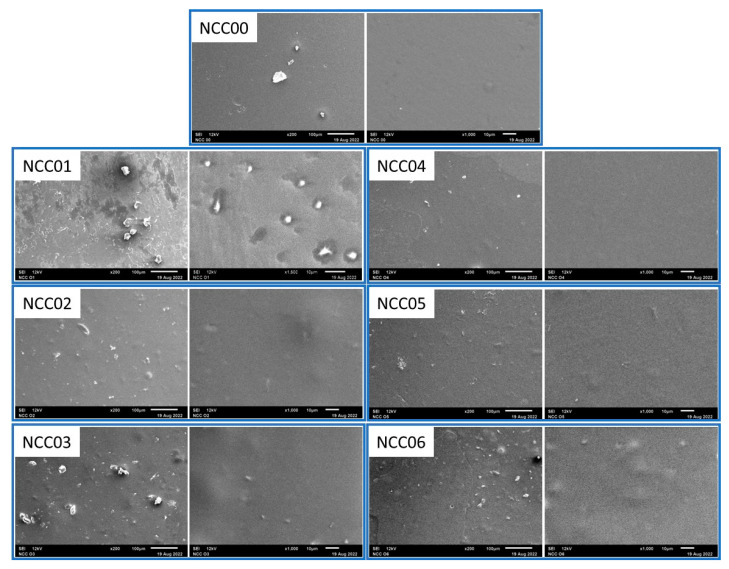
The surface SEM image of the prepared NEF samples.

**Figure 6 polymers-14-03738-f006:**
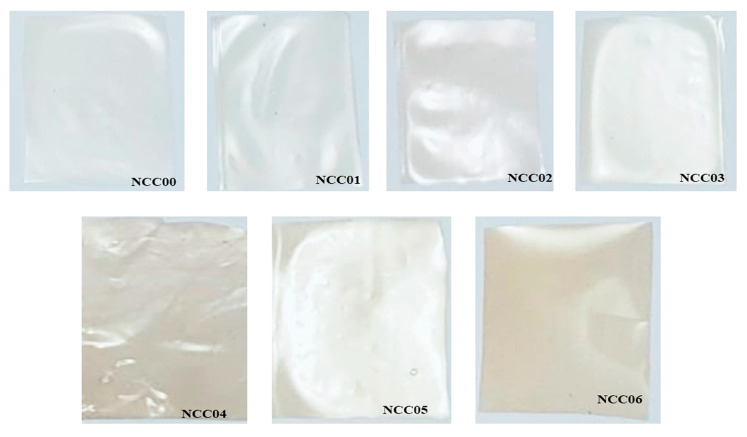
The photographs of the NEF samples.

**Table 1 polymers-14-03738-t001:** Composition of the bio-nanocomposites edible film prepared in this study.

Sample Code	Gelatin (% *w*/*v*)	Carboxymethyl Cellulose (% *w*/*v*)	Glycerol % (*v*/*v*)	Cellulose Nanocrystal (% *w*/*w*)	Nanopropolis (% *w*/*v*)
NCC00	1	1	0.25	-	-
NCC01	1	1	0.25	1	-
NCC02	1	1	0.25	3	-
NCC03	1	1	0.25	5	-
NCC04	1	1	0.25	1	0.1
NCC05	1	1	0.25	3	0.1
NCC06	1	1	0.25	5	0.1

**Table 2 polymers-14-03738-t002:** The summary of pH, water vapor permeability (WVP), oxygen permeability transmission rate (OPTR), and oxygen permeability (OP) of the prepared NEF samples.

Samples	pH	WVP × 10^7^ (g m^−1^ Pa^−1^ h^−1^)	OPTR (g m^−1^ d^−1^)	OP × 10^9^ (g m^−1^ Pa^−1^ d^−1^)
NCC00	6 ± 0.0	3.6 ± 0.2 ^b^	3.1 ± 0.2 ^b^	4.6 ± 0.3 ^b^
NCC01	6 ± 0.0	2.9 ± 0.1 ^a^	2.3 ± 1.1 ^ab^	3.1 ± 1.4 ^ab^
NCC02	6 ± 0.0	3.0 ± 0.2 ^a^	2.2 ± 0.6 ^ab^	3.2 ± 0.8 ^ab^
NCC03	6 ± 0.0	2.9 ± 0.3 ^a^	2.0 ± 1.7 ^ab^	3.1 ± 2.7 ^ab^
NCC04	6 ± 0.0	3.0 ± 0.3 ^ab^	1.6 ± 1.4 ^ab^	2.6 ± 2.3 ^ab^
NCC05	6 ± 0.0	3.4 ± 0.4 ^ab^	0.9 ± 0.1 ^a^	1.4 ± 0.2 ^a^
NCC06	6 ± 0.0	3.1 ± 0.3 ^ab^	1.0 ± 0.5 ^a^	1.5 ± 0.8 ^a^

Superscripts in the same column with the same letter show no significant differences in the Duncan’s test (α = 0.05).

**Table 3 polymers-14-03738-t003:** Glass transition temperature (Tg), melting temperature (Tm), and enthalpy.

Samples	Tg (°C)	Tm (°C)	Enthalpy (×10^3^ J/kg)
NCC00	127.76	325.11	44.20
NCC01	137.06	353.92	26.58
NCC02	130.34	354.68	76.16
NCC03	151.39	352.89	62.25
NCC04	137.24	325.58	42.92
NCC05	128.63	311.13	16.76
NCC06	131.79	304.40	77.19

**Table 4 polymers-14-03738-t004:** Transmittance at different wavelengths and transparency.

Samples	Transmittance (%) at a Different Wavelength (nm)						Transparency (mm^−1^)
	300	400	500	600	700	800	
NCC00	13.82 ± 2.06 ^b^	48.27 ± 2.03 ^e^	63.05 ± 1.58 ^d^	69.60 ± 1.21 ^d^	72.73 ± 1.00 ^d^	74.46 ± 0.81 ^d^	11.04 ± 0.20 ^a^
NCC01	12.61 ± 1.50 ^b^	43.31 ± 1.14 ^de^	57.45 ± 1.18 ^c^	64.29 ± 1.71 ^c^	68.00 ± 1.89 ^c^	70.28 ± 1.91 ^c^	12.49 ± 0.62 ^ab^
NCC02	11.85 ± 3.46 ^b^	41.76 ± 5.29 ^cd^	55.78 ± 3.98 ^bc^	62.57 ± 2.82 ^bc^	66.23 ± 2.12 ^bc^	68.49 ± 1.75 ^c^	13.09 ± 1.31 ^b^
NCC03	10.94 ± 2.42 ^b^	38.54 ± 3.46 ^bc^	51.56 ± 3.38 ^ab^	58.76 ± 1.70 ^a^	61.39 ± 2.88 ^a^	63.67 ± 2.54 ^a^	14.19 ± 0.73 ^c^
NCC04	6.32 ± 2.76 ^a^	36.81 ± 3.30 ^ab^	55.63 ± 1.40 ^bc^	64.54 ± 2.85 ^c^	67.99 ± 2.04 ^c^	70.43 ± 2.24 ^c^	12.21 ± 1.34 ^ab^
NCC05	4.59 ± 1.96 ^a^	32.66 ± 1.76 ^ab^	51.57 ± 0.80 ^ab^	60.01 ± 1.99 ^ab^	64.87 ± 2.41 ^ab^	67.92 ± 1.93 ^bc^	14.06 ± 0.66 ^c^
NCC06	3.91 ± 1.97 ^a^	30.91 ± 4.23 ^a^	49.87 ± 2.39 ^a^	57.94 ± 1.47 ^a^	62.56 ± 0.93 ^ab^	65.07 ± 0.81 ^ab^	14.48 ± 0.24 ^c^

Superscripts in the same column with the same letter show no significant differences in the Duncan’s test (*α* = 0.05).

**Table 5 polymers-14-03738-t005:** Color properties of NEF samples.

Samples	Δ*L**	Δ*a**	Δ*b**	Δ*E**
NCC00	16.079 ± 1.276 ^a^	2.368 ± 1.266 ^ab^	4.852 ± 0.725 ^d^	17.002 ± 1.293 ^a^
NCC01	18.990 ± 2.397 ^ab^	2.650 ± 0.838 ^b^	4.065 ± 0.711 ^cd^	19.624 ± 2.355 ^ab^
NCC02	18.897 ± 1.107 ^ab^	2.293 ± 0.026 ^ab^	2.501 ± 1.281 ^c^	19.221 ± 1.250 ^ab^
NCC03	19.340 ± 0.402 ^b^	1.983 ± 0.203 ^ab^	2.556 ± 0.758 ^c^	19.618 ± 0.485 ^ab^
NCC04	20.754 ± 2.490 ^bc^	1.290 ± 0.636 ^ab^	−3.565 ± 1.556 ^a^	21.139 ± 2.540 ^bc^
NCC05	23.573 ± 1.916 ^bc^	1.054 ± 0.394 ^a^	−4.177 ± 1.051 ^a^	23.974 ± 2.050 ^bc^
NCC06	20.761 ± 1.046 ^c^	1.132 ± 0.846 ^a^	−1.302 ± 1.611 ^b^	20.886 ± 1.023 ^c^

Superscripts in the same column with the same letter show no significant differences in the Duncan’s test (*α* = 0.05).

## Data Availability

The data presented in this study are available on request from the corresponding author.

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
