# Peer review of "Bio-Nanocomposite Based on Edible Gelatin Film as Active Packaging from Clarias gariepinus Fish Skin with the Addition of Cellulose Nanocrystalline and Nanopropolis"

_polymers, 2022, doi:10.3390/polym14183738_

Round 1

Reviewer 1 Report

This paper describes the synthesis and properties of biofilms that could be used as active packaging. The work itself is interesting. However, the authors extracted each of the main components: gelatin and nanocrystalline celulose. What worries me is the fact that the characterization of those materials was not provided. In fact, it is hard to believe that the CNC and gelatin were obtained without proper spectroscopic and microscopic characterization. In my opinion, without this information, this work is not suitable for publication in the Polymers journal.

Other comments:

-Some sentences in the Introduction are not clear and should be corrected, e.g. lines 54-56, line 60-61, line 83.

-In the preparation of CNC, what power and time were applied in the alkalization process?

-Line 198: 40 degrees per minute is quite fast. Is this correct?

-Line 438: High-quality SEM micrographs should be included. I can see that those pictures are not uniform in size (see NCC03 and NCC02). Moreover, when comparing, the same magnitude should be used - these are different for each sample.

Reviewer 2 Report

Cellulose nanocrystalline (CNC) and nanopropolis (NP) were incorporated in the mixed gelatin and cellulose films. Effects of CNC and NP concentrations on physical and mechanical properties of the composites were investigated. There are several issues about the quality of the research which are mainly concerned. These should be addressed before consideration.

-More discussion is required to differentiate effects of CNC and NP concentrations on properties of the composites.

- Carboxymethyl cellulose was used as a matrix of composites. It would be better to modify the title of the work.

-The term active packaging and edible film were used in the title. These should be clarified.

-The terms of “cellulose nanocrystal” and “cellulose nanocrystalline” were used in the manuscript, it would be better to constantly use only one term throughout the manuscript.

-A SEM or TEM image of CNC is required with a size and length.

-Line 29, Tensile strength and young's modulus increased significantly for the films loaded with >3% CNC. This sentence was misleading because no improvement in the tensile strength of the composites was observed with the addition of CNC.

-Line 232-233, when NGC was composited with NP, the thickness in fact decreases albeit a small decrease (comparing between NCC01-NCC03 with NCC04 and NCC05) before it slightly increases again (NCC06). From figure 2., with the incorporation of NP, the same letter was shown for the composites with the same amount of cellulose nanocrystalline. This implied no changes in thickness of the composites with the addition of NP.

-Line 280, The finding shows that the addition of CNC filler can reduce the water solubility of the resulting NEFs. The same superscript letter was provided for results of water solubility obtained for all materials except NCC01. This might explain the introduction of CNC and NP does not affect the water solubility of the composites.

-The ANOVA results showed no significant effects on WVP as demonstrated by P  > 0.05. The highest WVP value was found in the NCC001 treatment with 3.6 × 10-7 g-1 m-1 Pa- 1 h-1, while the lowest was in the NCC03 with 2.9 × 10-7 g-1 m-1 Pa-1 h-1. This sentence might mislead readers that NC003 had a better water vapor permeability than NC001, but the same statistical letter was provided, indicating the same level of the WVP performance.

-The following articles would be useful for discussion in the scope of the effect of CNC contents on mechanical properties of the composites.

https://onlinelibrary.wiley.com/doi/abs/10.1002/app.45710

https://onlinelibrary.wiley.com/doi/full/10.1002/app.45010

https://www.sciencedirect.com/science/article/pii/S0144861722000960

-The role of NP in composites should be further discussed in thermal and mechanical results.

-The quality of SEM images was poor. The SEM images with the better quality and higher magnification should be provided.

-Difference of transmittance and transparency should be explained.

-∆b*-value of NCC05 was higher than that of NCC06 and lower than that of NC004. Please explain this point.

-The photograph of NCC05 with less reflection should be presented, and effect of NP on color parameters of the composites should be pointed out.

- It should also discuss why NCC has a less reinforcing ability in the mixed gelatin-cellulose matrix in comparison to previous works.

-It would be better to modify the codes to match the CNC contents and the availability of NP such as 3CNCNP, the composite with 3 wt% of CNC and NP.  

Reviewer 3 Report

This manuscript reports the preparation and properties of edible gelatin film as packaging with the addition of cellulose nanocrystalline and nanopropolis. The experiments are well designed and the results are reliable. The manuscript is well organized and of interests to broad readers. The manuscript needs minor revision before acceptance for publication. My comments are as follows:

1.    It is better to revise the title as “Bio-nanocomposite Based on Edible Gelatin Film as Active Packaging from Clarias gariepinus Fish Skin with the Addition of Cellulose Nanocrystalline and Nanopropolis”.

2.    It is better to keep the same writing of “bio-nanocomposite” in the whole manuscript.

3.    Biomass materials are good precursors for food packing. Some related references are suggested to be cited in the introduction part, e.g. Packaging and degradability properties of polyvinyl alcohol/gelatin nanocomposite films filled water hyacinth cellulose nanocrystals; Molded fiber and pulp products as green and sustainable alternatives to plastics: A mini review; Application of Gelatin in Food Packaging: A Review; A review of cellulose and its derivatives in biopolymer-based for food packaging application.

4.    The authors should double check the format. Some words are in red and some are in different font (e.g. line 232 to 234). The language needs to be polished.

5.    It would be better to add TGA data to verify the heat resistance properties of NEF if possible. Please refer and cite “High Strength and Stable Proton Exchange Membrane Based on Perfluorosulfonic Acid/Polybenzimidazole” and “Electrospun Poly[poly(2,5-benzophenone)]bibenzopyrrolone/polyimide nanofiber membrane for high-temperature and strong-alkali supercapacitor”.

6.    The resolution of SEM images in Figure 5 is too low. SEM images with better definition should be offered.

7.    The photos if Figure 6 are indistinct. Better photos are suggested to be supplied.

Round 2

Reviewer 1 Report

The overall quality of the presented manuscript increased after the revisions. I recommend publishing it in the present form.

Author Response

Thank you for the valuable input from the reviewers to improve our manuscript quality.

Reviewer 2 Report

The manuscript has been improved, but the qaulity of SEM images shown in Fig 5 were not in the good shape for publication. Please take this comment seriously.

Author Response

Thank you for the follow-up comments.
We take seriously the reviewer's comment. We have conducted additional SEM analysis to obtain more representative images. Figure 5 in the revised manuscript contains a new set of sem images, two for each sample. we hope, the new data meet the standard set by the reviewer.

Additional discussion was included as well. See lines 451-460 and 478-480 (Figure 5). 

Round 3

Reviewer 2 Report

This revised manuscript is ready for publication!